# Temperature time series analysis at Yucatan using natural and horizontal visibility algorithms

**J. Alberto Rosales-Pérez**[1]*, **Efrain Canto-Lugo**[1], **David Valdés-Lozano**[2], **Rodrigo Huerta-Quintanilla**[1]

**1** Departamento de Física Aplicada, Centro de Investigación y de Estudios Avanzados del Instituto Politécnico Nacional. Unidad Mérida, Mérida, Yucatán 97310, México, **2** Departamento de Recursos del Mar, Centro de Investigación y de Estudios Avanzados del Instituto Politécnico Nacional. Unidad Mérida, Mérida, Yucatán 97310, México

* jose.alberto288148@gmail.com

## Abstract

Several methods to quantify the complexity of a time series have been proposed in the literature, which can be classified into three categories: structure/self-affinity, attractor in the phase space, and randomness. In 2009, Lacasa et al. proposed a new method for characterizing a time series called the natural visibility algorithm, which maps the data into a network. To further investigate the capabilities of this technique, in this work, we analyzed the monthly ambient temperature of 4 cities located in different climatic zones on the Peninsula of Yucatan, Mexico, using detrended fluctuation analysis (structure complexity), approximate entropy (randomness complexity) and the network approach. It was found that by measuring the complexity of the dynamics by structure or randomness, the magnitude was very similar between the cities in different climatic zones; however, by analyzing topological indices such as Laplacian energy and Shannon entropy to characterize networks, we found differences between those cities. With these results, we show that analysis using networks has considerable potential as a fourth way to quantify complexity and that it may be applied to more subtle complex systems such as physiological signals and their high impact on early warnings.

## Introduction

The variation in ambient temperature in a geographic zone is a combination of well-organized behavior as well as chaotic behavior in constant evolution and adaptation. The former is modulated by solar radiation and the latter by the diversity and distribution of vegetation and bodies of water around the planet, which makes a dynamical system more complex depending on how many factors affect the heating. All of this operates in diverse space and time scales through nonlinear interactions [1]. Ambient temperature is present in all processes involved not only in preserving life on the planet but also in degradation processes, such as materials

corrosion, which is a hot topic highly in coastal cities, such as those on the Yucatan Peninsula, where this research study was carried out [2], [3], [4], [5], [6]. Because the variation in temperature comes from a complex system [7], new analysis techniques have been developed and applied to obtain information that helps us understand its dynamics [8].

**Funding:** To all doctoral students, the Mexican government through CONACYT (which is the agency responsible for science and technology) gives us a monthly scholarship.

Complexity is one of the most important measurements needed to analyze time series such as temperature variation. Three approaches for measuring the complexity of a time series can be found in the literature: the characterization of attractors in the phase space, the self-affine structure of the time series and the degree of randomness. The first one can be approached by he Lyapunov exponent or Poincaré diagrams [9], [10]. The second one can be measured by detrended fluctuation analysis (DFA) or fractal dimension [11], [12]. The third one can be treated using approximate entropy (ApEnt) or sample entropy [1], [13].

An alternative way to analyze time series is by mapping their structure into a network. According to the information they obtain, algorithms can be divided into two groups: those that map similar dynamic states by drawing information from the phase space [14], [15], [16] and those that map the structure of the time series [17], [18].

The analysis of time series using networks is a method to characterize the dynamics of a system in an integral way. Lacasa [17] demonstrated that the structural and order/disorder characteristics of a dynamic system are extracted in the transformation. When the network is obtained, the connectivity can be characterized to analyze its structure, and its order/disorder can also be analyzed by measuring the entropy of its degree distribution. For the phase state attractor, there is no construction yet.

In addition to analyzing the structure and its order/disorder characteristics, we can analyze other features of the network that can be associated with the dynamics, such as the energy of a series, by computing the energy of the network.

Due to the simplicity of the algorithm, its low computational cost, and the structural information that can be obtained from the dynamics [19], the Lacasa et al. visibility algorithm has been very popular in diverse applications, such as the study of the time series of solar spots [20], the strength and frequency of hurricanes impacting the US [21], paleoclimatic and tidal measures [22], seismicity [23], [24], and finance [25], [26].

The aim of this article is to verify the hypothesis that with the network approach, we may obtain different insights from the analyzed dynamic system than those obtained from the previously mentioned techniques that focus on the structure or randomness of a time series. We used the climate as a study system, in particular the differences in the dynamics of cities that have distinct geographies but are relatively close to each other. We conducted our research in this way because this system has been studied extensively and is very well known, which allows us to conclusively verify which techniques work the best. For this purpose, we analyzed the ambient temperature time series in cities with different geographical characteristics, and we want to determine whether it is possible to differentiate between the dynamics of distinct geographical points that are located within a short distance from one another. We used DFA as a structural analysis technique and ApEnt as a randomness technique, which are famous tools used in nonlinear time series analysis.

## Materials and methods

### Transformation algorithms

In order to build the natural visibility algorithm (NVA) [17], we started with a uniformly sampled time series $y_t$: $t = 1, 2 \ldots N$, where each datapoint in the series represents a node. The connection between any two given datapoints $y_i$, $y_j$ will have visibility or will be linked (see Fig 1)

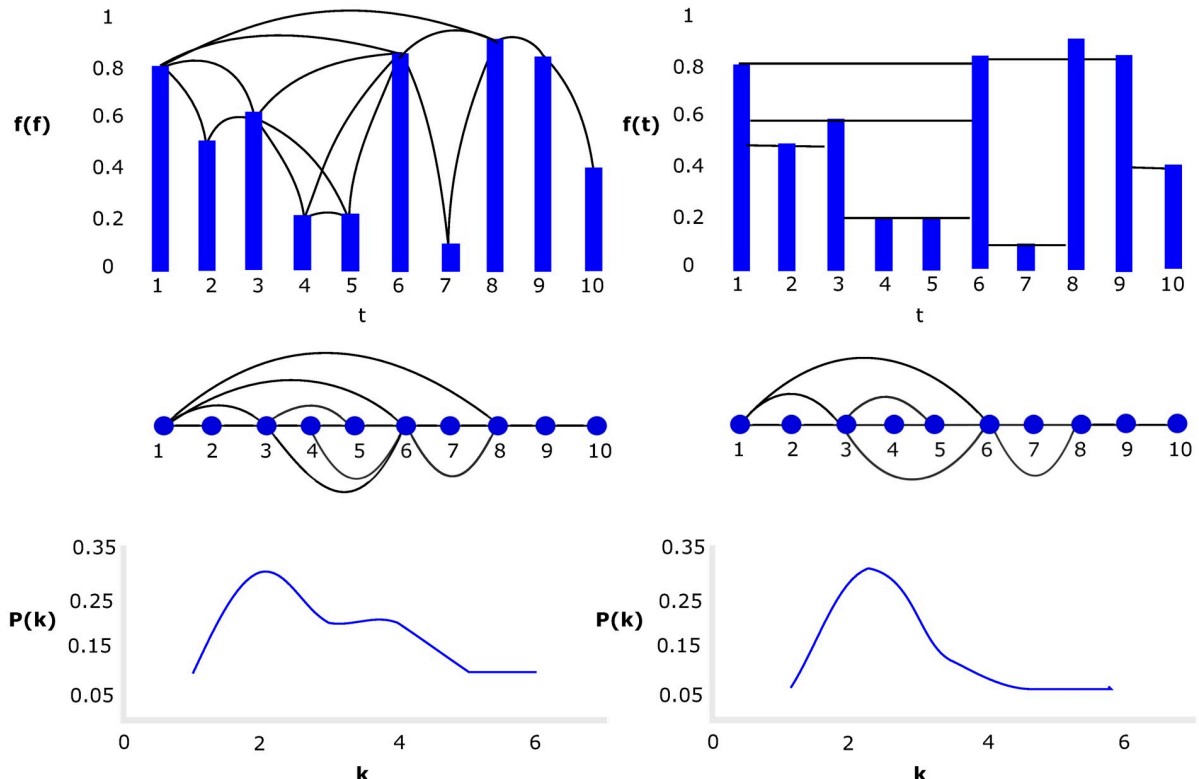

**Fig 1. From time series to networks.** Mapping a time series to a network using NVA (left) and HVA (right). The figure shows the time series with a solid line if there is visibility (link) between the indicated points. The generated network and its degree distribution are shown. It is observed that NVA has more connections than HVA by the visibility rules.

if any other point in the series $y_k$ found between $y_i$ and $y_j$ follows Eq (1):

$$\frac{y_j - y_k}{j - k} > \frac{y_j - y_i}{j - i} \tag{1}$$

There is a variant for the previous algorithm called the horizontal visibility algorithm (HVA) (Fig 1). This algorithm differs from the other algorithm in the condition on which the nodes $i$ and $j$ are linked, according to Eq (2):

$$y_i, y_j > y_k, \forall k | i < k < j \tag{2}$$

## Network characterization

Once the network has been generated, we characterize it with some metrics that will be presented in this section [27], [28]. Let $G$ be a network with $m$ links and $N$ nodes, where the number of nodes linked with node $i$ is called the degree of node $i$, denoted by $k_i$, and the mean degree is the average number of neighbors over all nodes in the network. To characterize the connectivity of the network, a histogram called the degree distribution is plotted. This diagram shows the probability $p(k)$ of taking a node with degree $k$, as shown in Fig 1.

The assortativity is a measure of the correlation of the network. If the most connected nodes prefer to be linked to the most connected nodes, the network is correlated, and its value is close to 1. On the other hand, if the less connected nodes prefer to link with the most

connected nodes, the network is anticorrelated, and its value is close to -1. A value of zero means that there is no preference at all.

Other measures have been suggested, such as Shannon entropy (Ent) and Laplacian energy (LE). The first one is defined as [29], where $k$ is the number of neighbors connected to the node and $p(k)$ is the probability of choosing a node with $k$ degree:

$$Ent(G) = -\sum_k p(k) ln(p(k)) \tag{3}$$

The LE calculation used for this study was generated by Kragujevac [30], [31], whose definition is:

$$LE(G) = 2m + \sum_{i=1}^{N} k_i^2 \tag{4}$$

where $k_i$ is the degree of the ith node. We have reported the Laplacian energy per node (LEN), which is $LE/N$.

## Data

A time series was obtained for the ambient temperature measured by weather stations located in the cities of Merida, Sierra Papacal, and Progreso, and Sisal (Fig 2) located in the state of Yucatan, Mexico from Jan 01, 2015, to Sep 28, 2016.

The data was recorded by the weather station DAVIS Vantage Pro 2 Plus [32]. Measurements were done every 2 seconds, and the average of all readings obtained within a 10 minute period was registered in the instrument's data base. Therefore, for each day, we will obtain 144 measurements. Each population was a group of 21 measurements, and we measured whether the means of each population are statistically different or not. Therefore, we measured the effectiveness of each parameter (DFA, ApEnt, and the network measures, such as mean degree, assortativity, Laplacian energy and Shannon entropy). The measuring equipment is owned by the CINVESTAV Department of Marine Resources, which is responsible for collecting the data and maintaining the equipment. The data acquisition process described previously is the way in which the equipment operates by default.

Data analysis was done on a monthly basis, out of which only the first 28 days were used to have time series with the same length. A general characterization was done with the mean and the standard deviation of the data. Afterwards, we applied DFA [11], [12] and ApEnt [1], [13]. On the other hand, the data was analyzed using networks, in which the time series for each month was transformed into a network using NVA and HVA. From each network, the following were obtained: degree distribution, mean degree, assortativity, Shannon entropy, and Laplacian energy per node.

A t-test was done for two independent means to determine whether the variables we are measuring are different, statistically speaking, among neighboring cities (see Fig 2), for example, Merida and Sierra Papacal, Sierra Papacal and Progreso, and so on. To determine whether two cities are different, we seek a p-value of $< 0.05$ in the t-test, which equals a 95% confidence. Each city will have a group of 21 independent and normally distributed measurements (DFA, ApEnt, mean degree, assortativity, Laplacian energy and Shannon entropy), one for each month.

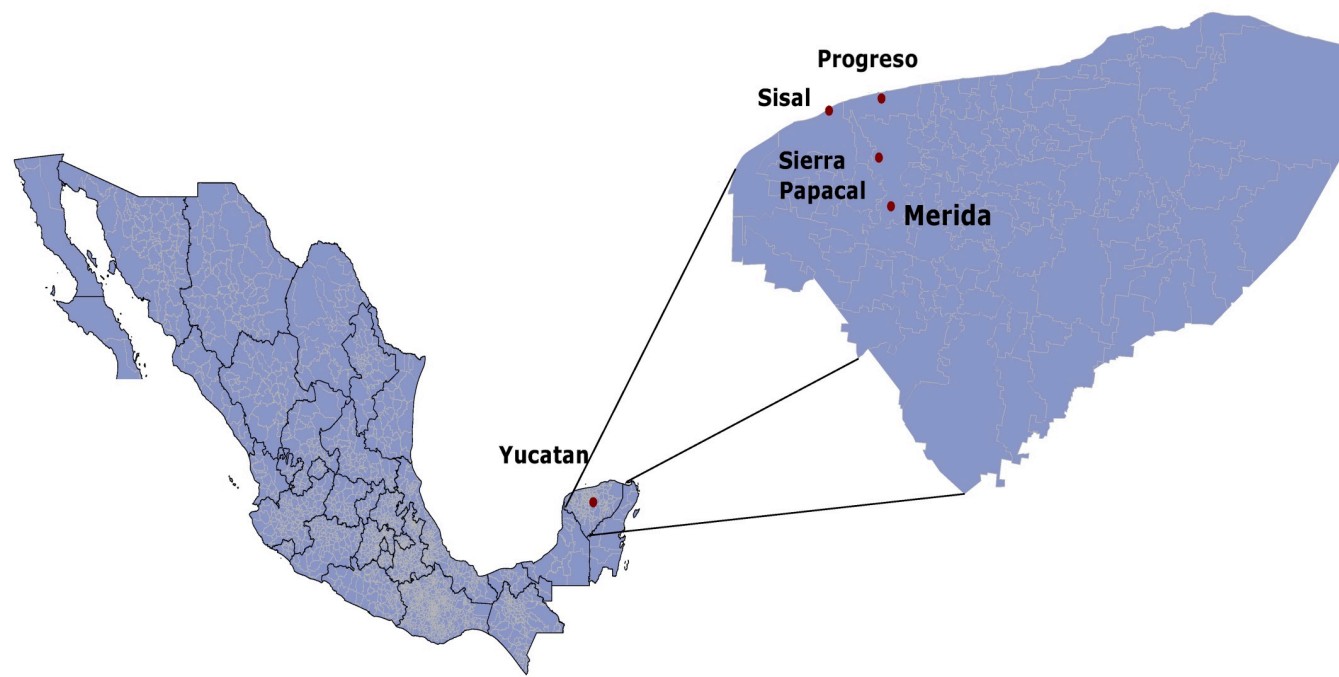

**Fig 2. Climate map of the state of Yucatan.** The map shows the locations of the measurements. Progreso and Sisal are coastal cities, whereas Sierra Papacal and Merida are not. According to the Köppen climate classification modified by García [33], Merida has an Aw2 climate (sub-humid warm), Sierra Papacal is BS1 (semi-arid very warm), and Progreso and Sisal are BS0 (arid very warm).

## Results

### Descriptive analysis

Table 1 shows the mean and the standard deviation of the data. Additionally, ApEnt and the DFA average coefficient in each city for the period between 2015 and 2016 are reported. It can be seen that the standard deviation is highest for Sierra Papacal, closely followed by Merida, and is lower in magnitude for the coastal cities Progreso and Sisal. For the ApEnt and DFA measurements, we found very similar values amongst the cities. For ApEnt, a differentiation could be made between the magnitudes for the coastal cities and Merida, with the magnitude of ApEnt for Sierra Papacal as an intermediate value. For the DFA value, we cannot make a very clear differentiation since all the values are close to 1.1; thus, the data correlation for all cities could be considered very similar.

In order to determine whether there is a statistically significant difference (p-value < 0.05, 95% confidence) between the means of two neighboring cities, for example Merida-Sierra Papacal (See the map in Fig 1), a t-test of the independent means was conducted (Table 2). It is

**Table 1. Descriptive statistics and complexity quantifiers for the cities analyzed in the Yucatan Peninsula.**

|  | Merida | Progreso | SierraP | Sisal |
|---|---|---|---|---|
| Mean(˚C) | 27.22 | 25.93 | 25.91 | 26.63 |
| SD | 3.33 | 2.34 | 3.54 | 2.61 |
| ApEnt | 0.2434 | 0.3114 | 0.2631 | 0.2931 |
| DFA | 1.14 | 1.10 | 1.12 | 1.09 |

**Table 2. P-values of the t-tests performed for neighboring cities.**

|  | ApEnt | DFA |
|---|---|---|
| Merida-SierraP | 0.051012 | 0.272336 |
| SierraP-Progreso | 0.00823* | 0.165988 |
| Progreso-Sisal | 0.207246 | 0.37817 |
| Merida-Progreso | 0.000897* | 0.073192 |

Values with an asterisk (*) represent a statistical significance of 95%.

noted that DFA cannot distinguish between cities, while ApEnt can differentiate between Sierra Papacal-Progreso and Merida-Progreso (non-coastal-coastal cities) but cannot differentiate between Merida and Sierra Papacal (non-coastal cities) or between Progreso and Sisal (coastal cities). This means that the complexity measured via ApEnt of the time series between coastal cities and between non-coastal cities is the same, although Merida and Sierra Papacal belong to different climatic zones.

## Network analysis

**NVA.** Table 3 shows the results of the network measures generated for each city. We found a downward trend in mean degree, LEN and Shannon entropy. There is no clear trend for assortativity.

Table 4 shows the t-test results. Mean degree, entropy, and energy are measures that can distinguish between the pairs Merida-Sierra Papacal and Sierra Papacal-Progreso. Furthermore, assortativity was not able to distinguish between these pairs. When the differences between coastal cities were analyzed, only the assortativity could differentiate between Progreso and Sisal.

Fig 3 shows the average degree distribution of the network generated for each month between 2015 and 2016. In the inset, the same data are shown but in a logarithmic scale. We can find that there is a bell in the middle of the degree distributions for the cities of Merida

**Table 3. Results for the networks generated by averaging the months in 2015 and 2016 per city.**

|  | Merida | Progreso | SierraP | Sisal |
|---|---|---|---|---|
| Mean Degree | 45.1 | 29.9 | 39.7 | 32.2 |
| Assortativity | -0.1145 | -0.0725 | -0.1193 | -0.1287 |
| Ent(G) | 1.99 | 1.86 | 1.95 | 1.88 |
| LEN(G) | 2919 | 1713 | 2397 | 1936 |

**Table 4. P-values of t-tests performed for neighboring cities for each metric in the networks presented in Table 3.**

|  | MeanDegree | Assortativity | Ent | LEN |
|---|---|---|---|---|
| Merida-SierraP | 0.000024* | 0.396552 | 0.000066* | 0.000037* |
| SierraP-Progreso | 0.00001* | 0.01132* | 0.00001* | 0.00001* |
| Progreso-Sisal | 0.057084 | 0.001293* | 0.104444 | 0.080656 |
| Merida-Progreso | 0.00001* | 0.00001* | 0.00001* | 0.00001* |

Values with an asterisk (*) represent a statistical significance of 95%.

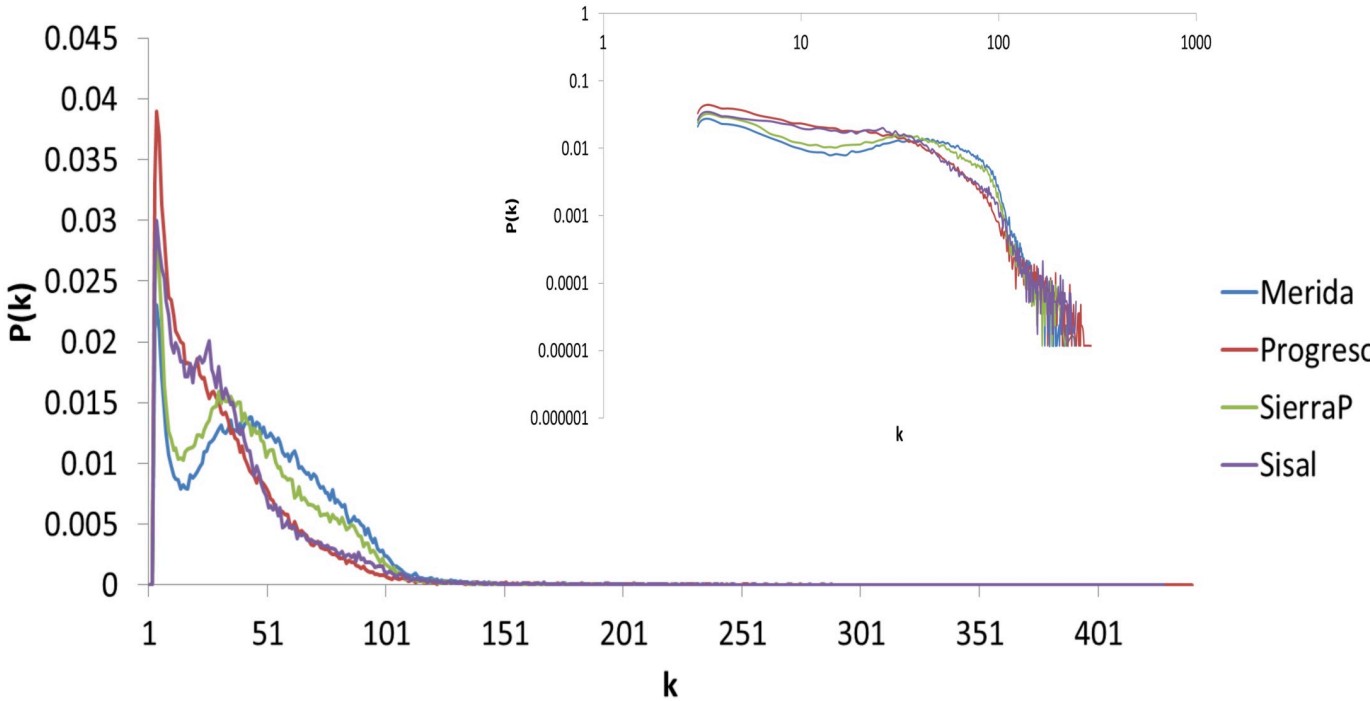

**Fig 3. Degree distributions for networks generated with NVA.** The figure shows the degree distributions in normal scales and in logarithmic scales in the inset. The non-coastal cites (Merida and Sierra Papacal) show a bell in the middle of the graph, while the coastal cities do not. These qualitative differences were quantified with the other connectivity measures mentioned above.

and Sierra Papacal, with maxima around k = 40 and k = 33, respectively. For Progreso and Sisal, this maximum is not as pronounced; instead, the values decrease smoothly until k = 100.

**HVA.** Table 5 shows the results for the transformation using HVA. We found that the tendency is not as shown in the NVA results. Among the cities, Sierra Papacal had the largest magnitudes of the mean degree and entropy and the smallest magnitude of the assortativity, contravening the NVA trend.

When the statistical differences (p-value <0.05, 95% confidence) between coastal cities *in* Table 6 *were analized*, we found a similar tendency for NVA except in the mean degree. Coastal cities and non-coastal cities can be differentiated amongst themselves, but coastal cities cannot.

Fig 4 shows the average degree distribution in the networks obtained with the transformation algorithm HVA. In general, the maximum occurs practically at the same *k*, and no clear tendency is observed.

The maximum of the degree distribution represents the number of average neighbors that a node will have. The higher the mean degree is, the more connections in the network. In this

**Table 5. Results for the networks generated by the averages of the months in 2015 and 2016 per city.**

|  | Merida | Progreso | SierraP | Sisal |
|---|---|---|---|---|
| Mean Degree | 3.09 | 2.97 | 3.16 | 3.03 |
| Assortativity | 0.1383 | 0.1834 | 0.0947 | 0.1585 |
| Ent(G) | 0.5898 | 0.5701 | 0.6100 | 0.5676 |
| LEN(G) | 14.19 | 13.24 | 14.99 | 21.35 |

**Table 6. P-values of t-tests performed for neighboring cities for each metric of the networks presented in Table 5.**

|  | MeanDegree | Assortativity | Ent | LEN |
|---|---|---|---|---|
| Merida-SierraP | 0.51075 | 0.004396* | 0.000178* | 0.000178* |
| SierraP-Progreso | 0.16361 | 0.00001* | 0.00001* | 0.00001* |
| Progreso-Sisal | 0.63892 | 0.134022 | 0.405995 | 0.153107 |
| Merida-Progreso | 0.00001* | 0.00001* | 0.00001* | 0.00001* |

Values with an asterisk (*) represent a statistical significance of 95%.

context, a time series without fluctuations (smooth) maps into a network with a higher connectivity than a series with variations (rough). As we can see in Fig 5, Merida is smoother than Progreso, and the maximum of its degree distribution using NVA is higher. This behavior is easy to see in the NVA but not in the HVA.

## Discussion

Since Merida is a city without bodies of water, its heating dynamics are similar to a plate under heating and cooling (like a sine wave), while in coastal cities, there are more factors that alter the ambient temperature, in this case, the sea and the swamp. In the middle is Sierra Papacal, which is a city that is between these two extremes: close to the effects of sea breezes but without bodies of water. In terms of factors that affect heating-cooling dynamics, if Merida is the least complex and Progreso is the most complex, Sierra Papacal would be an intermediate. DFA

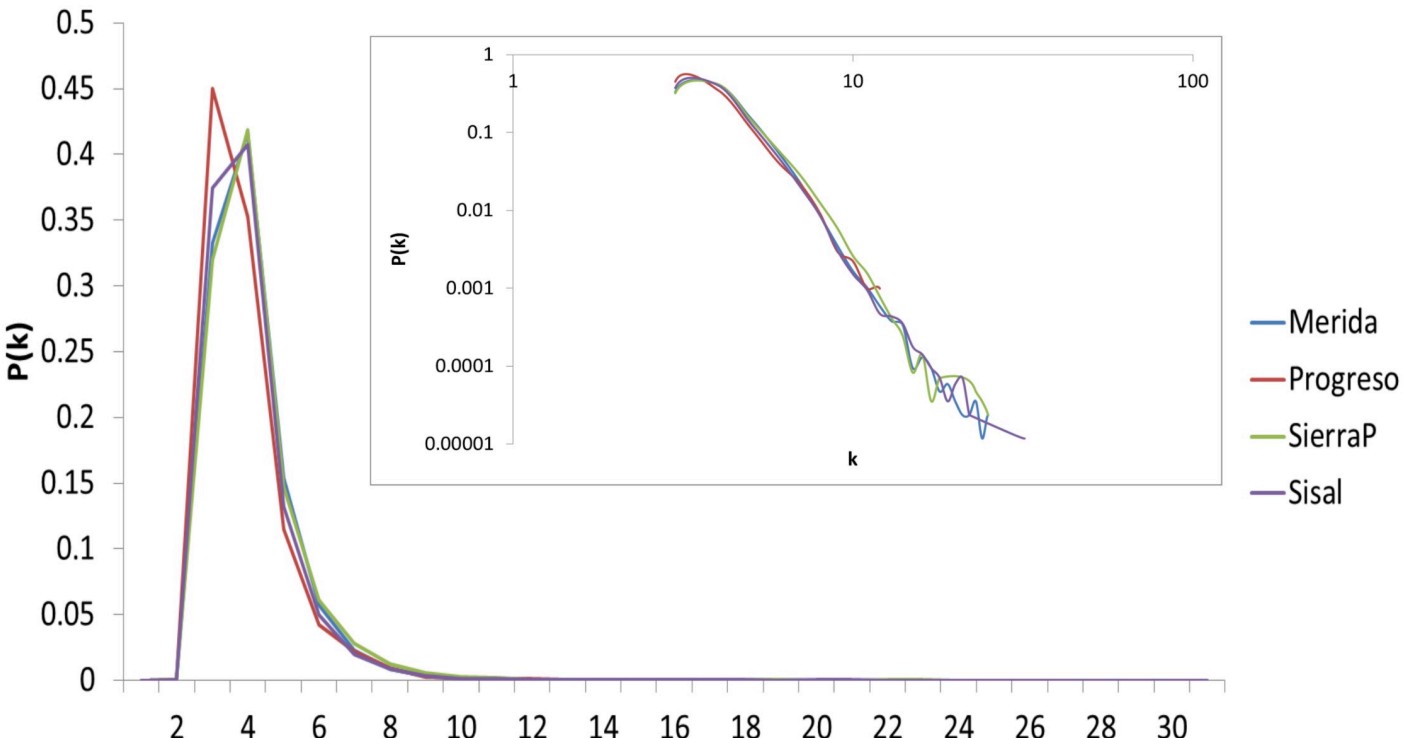

**Fig 4. Degree distribution of networks generated with HVA.** The degree distributions in normal scales and in logarithmic scales in the inset. None of the visual representations shown display any pattern that allows us to qualitatively determine any insights, as observed in the degree distributions generated by NVA.

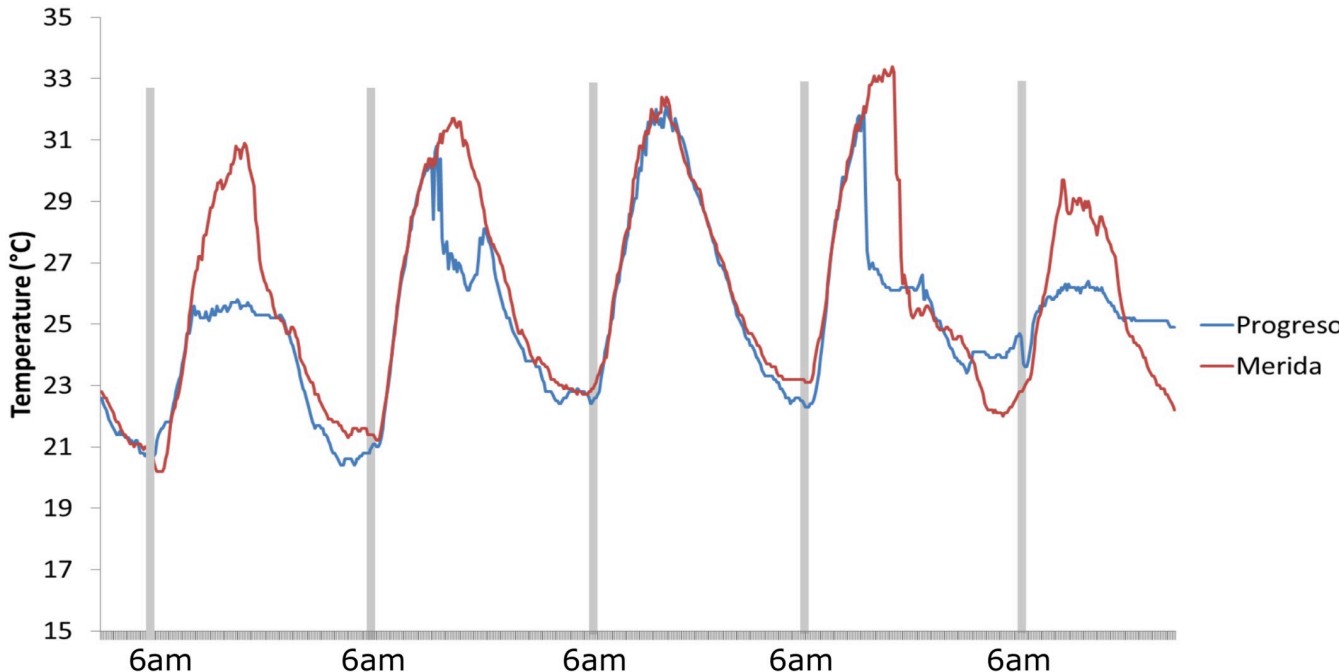

**Fig 5. Time series of Merida and Progreso, from January 1 to January 5, 2015.** The graph shows that Merida has a time series with fewer fluctuations than that of Progreso, and in this case, the fluctuation or "smoothness" of the data generates greater visibility between the datapoints. The more rugged the time series is, the lower the visibility and connectivity, and therefore, the mean degree will be lower.

and ApEnt could not differentiate between Merida and Sierra Papacal, generating the same magnitude of complexity for these two cities.

This cannot be true because Sierra Papacal is closer to the effects of sea breezes than Merida and is also in an intermediate climatic zone, according to the Köppen climate classification modified by García. With the networks, we find that we can differentiate the 3 cities with NVA and HVA and that these differences are statistically significant. Therefore, we can assign a measure of complexity using networks that conventional methods such as DFA or ApEnt cannot.

When the dynamics are analyzed by the network method, NVA not only statistically differentiates Sierra Papacal and Merida in both structure (mean degree) and randomness (Shannon entropy) but also tells us that there are differences between the energies of these dynamics (Laplacian energy). This finding agrees with the expected outcome since these two cities are in different climatic zones and shows that the analysis from the point of view of networks can be complementary to traditional techniques. This can be understood as a time series being analyzed using a Fourier transform, wherewe are "seeing" the same data but in the frequency space. This helps us nalyze properties that cannot be detected in the space of time. In the same way, when the Lacasa transformation algorithm is applied and we move to the network space, it allows us to measure properties of the dynamics that are not visible in the space of time.

If, on the one hand, we think of Merida as a city whose temperature dynamics are less complex because it does not have as many factors that affect it as Progreso, a city surrounded by water and, on the other hand, we observe using NVA that non-coastal cities generate networks with more connections, higher energy and higher entropy than coastal cities, then we can say that the complexity of a dynamic system is inversely proportional to the number of connections and the amount of energy and entropy in the generated network. This is to be expected

since a regular network has more connections than a random network (for example, an Erdos-Renyi model), so in this study, we showed that by using NVA, we can quantify the complexity of a dynamic system from the point of view of networks. However, the term complexity must be used very carefully, and further discussion is necessary in order to establish what complexity in the network space is.

When the networks are analyzed using HVA, we find that Sierra Papacal has higher magnitudes than Merida or Progreso. This result makes us think that this algorithm can detect something similar to a "phase transition" since Sierra Papacal is in an intermediate climatic zone between drought and humidity. We can consider Merida as a solid state and Progreso as a liquid state since the former is more regular than the latter and consider Sierra Papacal as the transition state. However, more studies have to be done to test this hypothesis.

## Conclusions

In this study, temperature time series in various cities with different geographies on the Yucatan Peninsula were characterized to further investigate the capabilities of this technique as an alternative to nonlinear analysis, such as DFA or ApEnt, for characterizing complexity.

We found that DFA and ApEnt cannot distinguish between coastal and non-coastal cities, although they belong to different climate zones, such Merida and Sierra Papacal, which are non-coastal cities but are in different climate zones. In contrast, the time series analysis using the NVA algorithm and characterization of the network through mean degree, entropy, and energy can distinguish between them. The other measurements showed poor performance.

When the NVA and HVA algorithms are compared, one finds that they generate very different tendencies, which leads to the hypothesis that each algorithm extracts different dynamic information: HVA extracts more subtle information and can quantify transitions states, whereas NVA characterizes the structure of the temperature time series, which depends on the geography of the city in which it was measured, and can be used as a complexity measure.

The findings mentioned in this paper allow us to establish the basis for continued work towards quantifying the complexity of a dynamic system through the space of networks. In this work, analysis using NVA and HVA was established as a holistic way of studying a dynamic system since such a system has the elements of structure and randomness and since other measurements such as energy can be integrated.

With more in-depth research, complexity quantifiers based on thermodynamics can be generated, which may be applied to more critical problems than climate classification, such as anticipating catastrophic events such as a stock market crash, a heart attack, an embolism or an epileptic attack.

## Supporting information

**S1 Appendix. Temperature dataset.** The document includes the temperatures for all cities analyzed in this study.
(CSV)

## Author Contributions

**Conceptualization:** J. Alberto Rosales-Pérez, Efrain Canto-Lugo, Rodrigo Huerta-Quintanilla.

**Data curation:** J. Alberto Rosales-Pérez, David Valdés-Lozano.

**Formal analysis:** J. Alberto Rosales-Pérez, Efrain Canto-Lugo.

**Investigation:** J. Alberto Rosales-Pérez, Efrain Canto-Lugo, David Valdés-Lozano, Rodrigo Huerta-Quintanilla.

**Methodology:** J. Alberto Rosales-Pérez, Efrain Canto-Lugo, Rodrigo Huerta-Quintanilla.

**Project administration:** Rodrigo Huerta-Quintanilla.

**Resources:** Efrain Canto-Lugo, David Valdés-Lozano, Rodrigo Huerta-Quintanilla.

**Software:** J. Alberto Rosales-Pérez, Efrain Canto-Lugo.

**Supervision:** Efrain Canto-Lugo, Rodrigo Huerta-Quintanilla.

**Validation:** J. Alberto Rosales-Pérez, Rodrigo Huerta-Quintanilla.

**Visualization:** J. Alberto Rosales-Pérez.

**Writing – original draft:** J. Alberto Rosales-Pérez.

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
