## [Decision Letter · Decision Letter 0]

5 Jul 2019

PONE-D-19-14940

Temperature Time series analysis at Yucatan using Natural and Horizontal Visibility Algorithms

PLOS ONE

Dear Mr Rosales Perez,

Thank you for submitting your manuscript to PLOS ONE. After careful consideration, we feel that it has merit but does not fully meet PLOS ONE’s publication criteria as it currently stands. Therefore, we invite you to submit a revised version of the manuscript that addresses the points raised during the review process.

All the reviewers raise several concerns regarding the methodology used in the paper. The authors need to carefully address all the detailed comments, and to clarify key issues regarding data availability, and significance of the experimental results. Among the major requested changes, the reviewers ask to better position the paper with respect to the existing approaches in the field, highlighting advantages and disadvantages with respect to long-established classification and clustering techniques. The time-scale used in the study should also be further discussed with regards to the experiments.

We would appreciate receiving your revised manuscript by Aug 19 2019 11:59PM. To enhance the reproducibility of your results, we recommend that if applicable you deposit your laboratory protocols in protocols.io, where a protocol can be assigned its own identifier (DOI) such that it can be cited independently in the future. For instructions see: http://journals.plos.org/plosone/s/submission-guidelines#loc-laboratory-protocols

We look forward to receiving your revised manuscript.

Kind regards,

Marco Lippi

Academic Editor

PLOS ONE

Journal Requirements:

2. Please amend your Financial disclosure statement to declare sources of funding, or state that the authors received no specific funding.

3. We note that Figure 2 in your submission contain [map/satellite] images which may be copyrighted. All PLOS content is published under the Creative Commons Attribution License (CC BY 4.0), which means that the manuscript, images, and Supporting Information files will be freely available online, and any third party is permitted to access, download, copy, distribute, and use these materials in any way, even commercially, with proper attribution. For these reasons, we cannot publish previously copyrighted maps or satellite images created using proprietary data, such as Google software (Google Maps, Street View, and Earth). For more information, see our copyright guidelines: http://journals.plos.org/plosone/s/licenses-and-copyright.

You may seek permission from the original copyright holder of Figure 2 to publish the content specifically under the CC BY 4.0 license. 

If you are unable to obtain permission from the original copyright holder to publish these figures under the CC BY 4.0 license or if the copyright holder’s requirements are incompatible with the CC BY 4.0 license, please either i) remove the figure or ii) supply a replacement figure that complies with the CC BY 4.0 license. Please check copyright information on all replacement figures and update the figure caption with source information. If applicable, please specify in the figure caption text when a figure is similar but not identical to the original image and is therefore for illustrative purposes only.

Please provide an amended Funding Statement that declares *all* the funding or sources of support received during this specific study (whether external or internal to your organization) as detailed online in our guide for authors at http://journals.plos.org/plosone/s/submit-now.  

Please state what role the funders took in the study.  If any authors received a salary from any of your funders, please state which authors and which funder. If the funders had no role, please state: "The funders had no role in study design, data collection and analysis, decision to publish, or preparation of the manuscript."

Additional Editor Comments:

All the reviewers raise several concerns regarding the methodology used in the paper. The authors need to carefully address all the detailed comments, and to clarify key issues regarding data availability, and significance of the experimental results. Among the major requested changes, the reviewers ask to better position the paper with respect to the existing approaches in the field, highlighting advantages and disadvantages with respect to long-established classification and clustering techniques. The time-scale used in the study should also be further discussed with regards to the experiments.

Reviewers' comments:

Reviewer's Responses to Questions

**Comments to the Author**

1. Is the manuscript technically sound, and do the data support the conclusions?

Reviewer #1: Partly

Reviewer #2: Yes

Reviewer #3: Partly

2. Has the statistical analysis been performed appropriately and rigorously? 

Reviewer #1: Yes

Reviewer #2: Yes

Reviewer #3: I Don't Know

3. Have the authors made all data underlying the findings in their manuscript fully available?

Reviewer #1: No

Reviewer #2: Yes

Reviewer #3: No

4. Is the manuscript presented in an intelligible fashion and written in standard English?

Reviewer #1: No

Reviewer #2: Yes

Reviewer #3: Yes

5. Review Comments to the Author

Reviewer #1: This paper explores the application of Lacasa's approaches to differentiate climate stations in southern Mexico. The application of the method is not very novel, but is technically sound. For this reason I recommend the paper be rejected. There is no rationale for why the study is being performed. For example -- what is the research problem? What is the difficulty in differentiating the climate regimes of weather stations in sub-tropical Mexico? What is your hypothesis?

Why was such a short time scale used in the study? it seems too short for climatology work. The effects of El Nino are overlooked in a short time span. Such regular longer-term variability can be greater than that explored on your study.

Reviewer #2: The authors use a different approach to analyze temperature series. I think it is a new tool in climate studies, but authors need to explain the findings more, and focus on what the network analysis offers that other statistical tools such as clustering cannot provide.

There are some parts that can be explained better to increase the quality of paper.

1. What kind of gain that can be observed by using HVA and NVA compared the any classification or clustering should be given clearly.

2. Why are data collected in every two minutes, and then averaged in 10 minutes period, and then analysis done on a monthly basis? It is not clear.

3. On page 6, first sentence mentions the bell max at k=40 and 33. How these values can be interpreted? What is the meaning of having k=40 or 33? Please give some feedback.

4. Why on the first 5 days of My 5, 2015 chosen to use the network?

5. On page 7, line 173, it is written that “NVA has more connections then HVA”. What is the advantage/disadvantage of having more connections?

6. On Figure 5 there is a star shape with 6 points for Merida. How one can interprets these 6 points? Also, there are few connections on the middle. What is the meaning of this?

7. On page 8 line 203 “my” should be “may”.

Reviewer #3: This paper is, I think, most naturally classified within the realm of "topological data analysis" (TDA), an area that I have been aware of for several years, though in my own case, only as an observer not as a participant. I have yet to see a paper in this field that, in my own view, provides a convincing analysis of an applied problem that could not have been achieved by more conventional methods, and this paper does not break that trend. Nevertheless I think the scientific literature should be open to new points of view even if they are not, initially, fully developed, and I feel the current paper should be publishable with some revisions intended mainly to clarify details of the method and to correct some minor errors.

My skepticism about the paper comes down essentially to this: there are established "linear" methods of time series analysis, such as autocorrelation plots, fitting autoregressive and moving-average models, spectral analysis and (if one slightly broadens the scope of the problem) multivariate methods such as principal components analysis and factor analysis, that could have been applied to address the problems in this paper, which essentially comes down to a classification problem distinguishing temperature time series at different locations in Yucatan. So if there is one "big picture" question I would like the authors to address in their revision it is this: why, in the authors' view, are the methods in the present paper superior to these long-established techniques?

Specific comments (using the 1-11 pages numbers on the authors' own copy):

Page 1, bottom: climate or weather? It seems to me this paper is primarily addressing daily weather patterns in different regions of Yucatan, and I don't see any implication for long-term trends (e.g. whether trends are greater on the coast than inland) following directly from this analysis. Terminology is important, and so is thinking about the broader implications of your work, so if you do see such implications, I would encourage you to develop them.

Page 2, line 20, DFA is introduced here but not explained, whereas later you write Detrended Fluctuation Analysis. My advice would be to spell out an acronym the first time you use it, but thereafter, once the meaning is established, writing DFA (and other acronyms used in this paper) would be fine

Page 2, line 37: the inserted word "it" is redundant. (But my broader concern is not with minor linguistic detail but the broader implication of this sentence: it seems to me you have not considered well-established time series techniques among the "other techniques" that you discuss.)

Page 2, line 46: Natural not NAtural

Page 2, lines 47-49: I would encourage the authors to be careful about consistency of notation. Why introduce the time series as y_t and then immediately switch to y_i and y_j?

Page 2, equation (1.1): there is something mysterious about how this formula appears and the way it is depicted in figure 1. On the face of it, temperature time series have both positive and negative values and there is nothing special about "zero" whether they are being measure in degrees Fahrenheit or Celsius (of course zero Kelvin does have a special physical interpretation, but weather time series never get anywhere close to that boundary). Formula (1) seems a little odd because if you reversed the signs the condition would change, and correspondingly, the picture in figure 1 would look different if you chose a different base value for zero. Possibly this has been previously discussed in the previous literature on these techniques, but it seems to me the classification would change if you reversed all the signs, and that with temperatures, somehow this should not happen. Any comment on these issues?

Page 3, lines 72 and 76: should be no indentation (if the manuscript was prepared using Latex then this is a common Latex error)

Page 4, top: please include references to K\\"oppen and Garc\\'ia

Page 4, around line 90: this is another general point of presentation that other reviewers might express differently, but my advice is that for methods that you actually use in your analysis, that here include DFA and ApEnt, you should provide enough information in the form of formulas or explicit references (one or the other) so that the reader who wants to repeat your analysis can reproduce it if so desired, whereas for other methods that you only mention in passing, such as Shannon Entropy, it's not necessary to be so explicit.

Page 4, around line 95: please clarify exactly how you are computing the t-test. My first reaction was that since you are looking at correlated time series, any use of t-tests or similar methods should include a correction for autocorrelation, but then I realized that if you are using a t-test to compare measures computed from widely separated blocks of the time series, maybe this kind of correction is not necessary. In nay case, I feel this point deserves some explicit discussion, i.e. either include a correction for autocorrelation or explain why it is not necessary.

Page 4, line 105: when you say that Merida shows a "significant" difference (but Sierra Papacal, for example, does not) you should state exactly what numerical measure you are using to judge what is "significant"

Page 4, lines 107-109: I am still looking for a meteorological interpretation of these results. It seems to be that, very broadly, what you are measuring is smoothness of the diurnal variation, and it is a general phenomenon that temperatures on the coast show less diurnal variation that those inland. Is this what is going on, or should we be thinking about some more complex interpretation?

Page 4, line 110: "very similar values". Same comment as the one about significance in Merida: I would like to know exactly what the numerical values were and how you judged that they either were or were not significantly different for the different cities.

Page 4, line 117: same comment as above about the "paired t-test": Please explain in a little more detail about what this was and whether autocorrelation is an issue in the way the t-test is calculated

Page 4, line 119, "all measures except the mean and DFA". Slightly strange wording here: it would be more straightforward to list the measures that were different than those that were not, in other words, say that SD and ApEnt showed a significant difference between these two cities

Page 5, line 130: magnitude

Page 6, lines 152-154: as with earlier comments about how you assess differences among cities, I would like you to be a little more explicit here, how exactly are you making these judgments.

Page 7, caption to Figure 5: I have to say that the interpretation of the plots is becoming a little harder for me as the paper goes on. Figure 5 has a pleasing geometric appearance but I am not at all sure how to interpret it. I think the relevant scientific question is this: are the results dependent on the specific choice of a 5-day window or are the authors in a position to state that there are some general patterns emerging in these plots that are invariant to irrelevant details such a which specific 5 days we chose for the analysis? I didn't see that question discussed.

Page 7, caption ot Figure 6: please check the wording of this caption for typos (repeated "it is") and Progreso is misspelled at one point.

Page 9, lines 255-257, English please!

Figure 4: could you please explain why the density plot appears to go below zero at around k=2?

On data availability: Note that the time series are in ten-minute intervals and the standard sources for meteorological data (e.g. NOAA or NASA in the USA) only list daily or monthly time series. I think the authors could give more explicit detail about the original source for the data (e.g. did they come from the Mexican meteorological service?) and how the data were prepared for the present application - if the original source of the data was public then there is no issue about data confidentiality.

Overall judgment: my principal concern about this paper is the one I expressed at the beginning, that the authors don't really explain why they need these particular techniques against far better established methods from the time series literature, but if they can clear up the various ambiguities about what they were actually doing, I would be willing to see this paper published in PLoS ONE.

6. PLOS authors have the option to publish the peer review history of their article (what does this mean?). If published, this will include your full peer review and any attached files.

Reviewer #1: No

Reviewer #2: No

Reviewer #3: No

---

## [Author Response · Author response to Decision Letter 0]

4 Sep 2019

Dear Reviewers and Editor

We appreciate your comments and suggestions. They helped us a lot in order to improve the article and also to be able to give a clearer understanding of the idea that we want to communicate to the readers. We hope that with these corrections the points we want to release will become clearer. We thank you for your time and patience.

The letter is organized as follows: First is the response to the editor and then the response to the 3 reviewers. Each answer is a letter for each one. We include the editor's comment and the answer begins with an A. 

Once again we thank you for your support. We look forward to your comments

Academic Editor

Thanks for your comments, here are the answers:

Journal Requirements:

A. The article was made with the temple of latex that we downloaded from the page, we reviewed the format and we attended your request to review and change what was not in the format.

2. Please amend your Financial disclosure statement to declare sources of funding, or state that the authors received no specific funding.

A. At this point I have doubt, the situation is as follows

Direct financial support was not received for the studio. To all doctoral students, the Mexican government through CONACYT (which is the agency responsible for science and technology) gives us a monthly scholarship. I understand that this financial disclosure is for direct support to the study and this scholarship is not intended for that, so I do not know if this scholarship enters the financial disclosure and how would it enter?

3. We note that Figure 2 in your submission contain [map/satellite] images which may be copyrighted. All PLOS content is published under the Creative Commons Attribution License (CC BY 4.0), which means that the manuscript, images, and Supporting Information files will be freely available online, and any third party is permitted to access, download, copy, distribute, and use these materials in any way, even commercially, with proper attribution. For these reasons, we cannot publish previously copyrighted maps or satellite images created using proprietary data, such as Google software (Google Maps, Street View, and Earth). For more information, see our copyright guidelines: http://journals.plos.org/plosone/s/licenses-and-copyright.

A. We changed the figure to one made by us to avoid situations with copyright

A. At this point I have doubt, the situation is as follows

Direct financial support was not received for the studio. To all doctoral students, the Mexican government through CONACYT (which is the agency responsible for science and technology) gives us a monthly scholarship. I understand that this financial disclosure is for direct support to the study and this scholarship is not intended for that, so I do not know if this scholarship enters the financial disclosure and how would it enter?

A. The data will be public

Additional Editor Comments:

All the reviewers raise several concerns regarding the methodology used in the paper. The authors need to carefully address all the detailed comments, and to clarify key issues regarding data availability, and significance of the experimental results. Among the major requested changes, the reviewers ask to better position the paper with respect to the existing approaches in the field, highlighting advantages and disadvantages with respect to long-established classification and clustering techniques. The time-scale used in the study should also be further discussed with regards to the experiments.

About these concerns, here is our answer:

A. What is pursued in this study is to generate the basics to analyze the dynamics from a more holistic point of view such as its transformation to networks. The objective of this article is not to use NVA and HVA as techniques that replace the ways in which climate is classified, but to be complementary by analyzing the problem from another point of view: the complexity of the dynamics through networks analysis. 

The objective of the article is to show if analyzing the dynamics as networks can generate different insights from those of other techniques such as DFA and ApEnt wich are considered basic techniques to measure complexity. We used climate as a study system, in particular the quantitative differentiation between the dynamics of cities with different geographies but relatively close to each other. We did it in this way since this system is very well known, which allows us to check without a doubt which techniques work best. We found that HVA and NVA can distinguish between geographically close dynamics (Merida vs Sierra Papacal) while DFA and ApEnt do not, which proves the hypothesis that analyzing time series using networks can generate complementary results than traditional techniques.

An approach to characterize dynamic systems is based on their complexity, which can be measured by:

• Structure or Self-Affinity, quantified by fractal dimension analysis, DFA

• Attractor in phase space, quantified by exponents of lyapunov, correlation dimensions

• Order / Disorderer state, studied using entropy and its different definitions (ApEnt, SampEnt)

These 3 ways of analyze complexity and measure it are interdependent and interrelated, however, each one focuses on a main characteristic of dynamics.

We do not contemplate the long-established classification and clustering techniques because these techniques are not used to measure complexity. Certainly the meaning of the article seems to be more a way of classifying and for this you can use these techniques, but the aim of the article is to establish the basis for measuring the complexity of the signals from of the analysis of the generated networks.

Lacasa in his article showed that the mapping with NVA and HVA can extract the structural and order / disorder characteristics of a dynamic. When the network is obtained, its connectivity can be characterized to analyze its structure (similar to self-affinity and fractal techniques) and we can analyze its order / disorder through the entropy of its degree distribution (similar to SampEnt and ApEnt). For the phase state attractor a simile has not yet been found, which would be a future job.

In addition to analyzing the structure and its order / order, we can analyze other characteristics of the dynamics such as the energy of a time series (measured using the Laplacian energy of the network). The analysis of this topological quantifier opens the door to integrate a framework based on thermodynamics, which we are also working on.

When we test the methods, we find that if we analyze the dynamics from the point of view of their structure / self-affinity (DFA) or by order / disorder (ApEnt), these methods could not differentiate between climates (Merida and Sierra Papacal), while that HVA and NVA can, which is a great INITIAL result, which proves to us the hypothesis that when analyzing the dynamics through the space of networks that integrates both the structure and the order / Disorder and the extra feature such as energy, We can find things that with traditional methods of complexity analysis cannot. 

You may wonder and where complexity connects with the differentiation of climates? Since Merida is a system without bodies of water, its heating dynamics is like a plate under heating, while in coastal cities there are more factors that alter the ambient temperature, in this case the sea and the swamp, which makes it a system more complex. In the middle is Sierra Papacal, which is a city that is between these two extremes, close to the effects of sea breezes but without bodies of water. We think that if Mérida is the least complex and Progreso is the most complex from the point of view of factors that affect the heating-cooling dynamics, Sierra Papacal would be an intermediate. DFA and ApEnt, could not differentiate between Mérida and Sierra Papacal, generating the same magnitude of complexity for these two cities, which cannot be because Sierra Papacal is closer to the effects of sea breezes than Mérida and is also in a intermediate climatic zone between Progreso and Merida according to the Köpen climate classification modified by García. What we find with the networks is that 1) if we can differentiate with NVA and HVA the 3 cities and these differences are statistically significant, so we can assign a measure of complexity across the space of the networks that conventional methods such as DFA or ApEnt do not .

Another great result is that we observe that HVA and NVA extract different information from the dynamics. In the literature, an in-depth analysis of what information is extracted by one algorithm or another has not been reported. What we find is that NVA has an upward or downward trend depending on which topological index is being analyzed when you measure the dynamics from the coast to land inside. On the other hand, HVA always highlights the city that lies between the coast and inland, and that in the allocation of Köpen climate zones Modified by Garcia, is an intermediate climate between wet and dry.

Since the assignment of links for HVA is much more restrictive than NVA because of how the algorithms are constructed, we hypothesize that HVA extracts more subtle information than NVA, so that the first one is much more sensitive and allows us to detect states of transition in complex systems. NVA, on the other hand, is able to extract more general trends from the dynamic. Which will also be addressed using different computational experiments in another article.

In summary, this study is focused on generate the basics for a more comprehensive exploration of the quantification of the complexity of a dynamic. The time series of ambient temperature of different cities was taken as an example since they come from a complex system which has been widely studied and whose results we can rely on to rate the effectiveness of the methods. In particular, the test focused on the statistically significant distinction of the dynamics analyzed by several methods, HVA and NVA being better than DFA, ApEnt.

In addition to the results obtained in this initial experiment, many lines of research were generated that are being worked on or that can be taken up by other researchers in such a way that a different twist is given to the analysis of time series using the mapping to networks, going from being merely descriptive to being an analysis that can measure the complexity of a dynamic in a robust way and can be applied with great certainty in other systems more difficult to understand as physiological signals.

We do not talk about the fact that we are measuring the complexity of a dynamic because it is a definition that must be taken very carefully and the approach of analyzing time series using networks with a complexity approach is just beginning to mature. We have corrected the article (introduction and discussion) in such a way that it can be understood as it is an exploratory analysis of complexity of a dynamics through the “space of networks” applied to the climate and not in itself a purely climatic analysis.

Reviewer 1

This paper explores the application of Lacasa's approaches to differentiate climate stations in southern Mexico. The application of the method is not very novel, but is technically sound. For this reason I recommend the paper be rejected. There is no rationale for why the study is being performed. For example -- what is the research problem? What is the difficulty in differentiating the climate regimes of weather stations in sub-tropical Mexico? What is your hypothesis?

Why was such a short time scale used in the study? it seems too short for climatology work. The effects of El Nino are overlooked in a short time span. Such regular longer-term variability can be greater than that explored on your study.

Dear reviewer, 

Thank you very much for your valuable comments; we see that we had misunderstood the meaning of the article. Let us clarify it:

The objective of this article is not to use NVA and HVA as techniques that replace the ways in which climate is classified, but to be generate the basics analyzing the problem from another point of view: the complexity of the dynamics through networks analysis. 

We want to show if analyzing the dynamics as networks can generate different insights from those of other techniques such as DFA and ApEnt wich are considered basic techniques to measure complexity. We used climate as a study system, in particular the quantitative differentiation between the dynamics of cities with different geographies but relatively close to each other. We did it in this way since this system is very well known, which allows us to check without a doubt which techniques work best. We found that HVA and NVA can distinguish between geographically close dynamics (Merida vs Sierra Papacal) while DFA and ApEnt do not, which proves the hypothesis that analyzing time series using networks can generate complementary results than traditional techniques wich quantifies complexity.

An approach to characterize dynamic systems is based on their complexity, which can be measured by:

• Structure or Self-Affinity, quantified by fractal dimension analysis, DFA

• Attractor in phase space, quantified by exponents of lyapunov, correlation dimensions

• Order / Disorderer state, studied using entropy and its different definitions (ApEnt, SampEnt)

These 3 ways of analyze complexity and measure it are interdependent and interrelated, however, each one focuses on a main characteristic of dynamics.

We do not contemplate the long-established classification and clustering techniques because these techniques are not used to measure complexity. Certainly the meaning of the article seems to be more a way of classifying and for this you can use these techniques, but the aim of the article is to establish the basis for measuring the complexity of the signals from of the analysis of the generated networks.

Lacasa in his article showed that the mapping with NVA and HVA can extract the structural and order / disorder characteristics of a dynamic. When the network is obtained, its connectivity can be characterized to analyze its structure (similar to self-affinity and fractal techniques) and we can analyze its order / disorder through the entropy of its degree distribution (similar to SampEnt and ApEnt). For the phase state attractor a simile has not yet been found, which would be a future job.

When we test the methods, we find that if we analyze the dynamics from the point of view of their structure / self-affinity (DFA) or by order / disorder (ApEnt), these methods could not differentiate between different climates zones (Merida and Sierra Papacal), while that HVA and NVA can, which is a great INITIAL result, which proves to us the hypothesis that when analyzing the dynamics through the space of networks. We can find things that with traditional methods of complexity analysis cannot. 

You may wonder, Where do complexity connects with the differentiation of climates? Since Mrida is a system without bodies of water, its heating dynamics is like a plate under heating, while in coastal cities there are more factors that alter the ambient temperature, in this case the sea and the swamp, which makes it a system more complex. In the middle is Sierra Papacal, which is a city that is between these two extremes, close to the effects of sea breezes but without bodies of water. We think that if Mérida is the least complex and Progreso is the most complex from the point of view of factors that affect the heating-cooling dynamics, Sierra Papacal would be an intermediate. DFA and ApEnt, could not differentiate between Mérida and Sierra Papacal, generating the same magnitude of complexity for these two cities, which cannot be because Sierra Papacal is closer to the effects of sea breezes than Mérida and is also in a intermediate climatic zone between Progreso and Merida according to the Köpen climate classification modified by García. What we find with the networks is that 1) if we can differentiate with NVA and HVA the 3 cities and these differences are statistically significant, so we can assign a measure of complexity across the space of the networks that conventional methods such as DFA or ApEnt do not .

Another great result is that we observe that HVA and NVA extract different information from the dynamics. In the literature, an in-depth analysis of what information is extracted by one algorithm or another has not been reported. What we find is that NVA has an upward or downward trend depending on which topological index is being analyzed when you measure the dynamics from the coast to land inside. On the other hand, HVA always highlights the city that lies between the coast and inland, and that in the allocation of Köpen climate zones Modified by Garcia, is an intermediate climate between wet and dry.

Since the assignment of links for HVA is much more restrictive than NVA because of how the algorithms are constructed, we hypothesize that HVA extracts more subtle information than NVA, so that the first one is much more sensitive and allows us to detect states of transition in complex systems. NVA, on the other hand, is able to extract more general trends from the dynamic. Which will also be addressed using different computational experiments in another article.

In summary, this study is focused on generate the basics for a more comprehensive exploration of the quantification of the complexity of a dynamic. The time series of ambient temperature of different cities was taken as an example since they come from a complex system which has been widely studied and whose results we can rely on to rate the effectiveness of the methods. In particular, the test focused on the statistically significant distinction of the dynamics analyzed by several methods, HVA and NVA being better than DFA, ApEnt.

In addition to the results obtained in this initial experiment, many lines of research were generated that are being worked on or that can be taken up by other researchers in such a way that a different twist is given to the analysis of time series using the mapping to networks, going from being merely descriptive to being an analysis that can measure the complexity of a dynamic in a robust way and can be applied with great certainty in other systems more difficult to understand as physiological signals.

We do not talk about the fact that we are measuring the complexity of a dynamic because it is a definition that must be taken very carefully and the approach of analyzing time series using networks with a complexity approach is just beginning to mature. We have corrected the article (introduction and discussion) in such a way that it can be understood as it is an exploratory analysis of complexity of a dynamics through the “space of networks” applied to the climate, and not in itself a purely climatic analysis, because, as you mentioned, more data is needed to take into account effects such as El Niño, La Niña, etc. Another study would be to know if the results obtained in this study are maintained or at large scales. We expect to be maintained since one of the qualities of complex systems is its scale invariance, however this has to be verified through a study.

We look forward to your comments, we appreciate your time

Reviewer 2:

Thank you very much for your comments, we see that there were several details that helped us to better support the article. Below I answer each comment:

The authors use a different approach to analyze temperature series. I think it is a new tool in climate studies, but authors need to explain the findings more, and focus on what the network analysis offers that other statistical tools such as clustering cannot provide.

There are some parts that can be explained better to increase the quality of paper.

1. What kind of gain that can be observed by using HVA and NVA compared the any classification or clustering should be given clearly.

What is pursued in this study is to generate the basics to analyze the dynamics from a more holistic point of view such as its transformation to networks. The objective of this article is not to use NVA and HVA as techniques that replace the ways in which climate is classified, but to be complementary by analyzing the problem from another point of view: the complexity of the dynamics through networks analysis. 

The objective of the article is to show if analyzing the dynamics as networks can generate different insights from those of other techniques such as DFA and ApEnt wich are considered basic techniques to measure complexity. We used climate as a study system, in particular the quantitative differentiation between the dynamics of cities with different geographies but relatively close to each other. We did it in this way since this system is very well known, which allows us to check without a doubt which techniques work best. We found that HVA and NVA can distinguish between geographically close dynamics (Merida vs Sierra Papacal) while DFA and ApEnt do not, which proves the hypothesis that analyzing time series using networks can generate complementary results than traditional techniques.

An approach to characterize dynamic systems is based on their complexity, which can be measured by:

• Structure or Self-Affinity, quantified by fractal dimension analysis, DFA

• Attractor in phase space, quantified by exponents of lyapunov, correlation dimensions

• Order / Disorderer state, studied using entropy and its different definitions (ApEnt, SampEnt)

These 3 ways of analyze complexity and measure it are interdependent and interrelated, however, each one focuses on a main characteristic of dynamics.

We do not contemplate the long-established classification and clustering techniques because these techniques are not used to measure complexity. Certainly the meaning of the article seems to be more a way of classifying and for this you can use these techniques, but the aim of the article is to establish the basis for measuring the complexity of the signals from of the analysis of the generated networks.

Lacasa in his article showed that the mapping with NVA and HVA can extract the structural and order / disorder characteristics of a dynamic. When the network is obtained, its connectivity can be characterized to analyze its structure (similar to self-affinity and fractal techniques) and we can analyze its order / disorder through the entropy of its degree distribution (similar to SampEnt and ApEnt). For the phase state attractor a simile has not yet been found, which would be a future job.

In addition to analyzing the structure and its order / order, we can analyze other characteristics of the dynamics such as the energy of a time series (measured using the Laplacian energy of the network). The analysis of this topological quantifier opens the door to integrate a framework based on thermodynamics, which we are also working on.

When we test the methods, we find that if we analyze the dynamics from the point of view of their structure / self-affinity (DFA) or by order / disorder (ApEnt), these methods could not differentiate between climates (Merida and Sierra Papacal), while that HVA and NVA can, which is a great INITIAL result, which proves to us the hypothesis that when analyzing the dynamics through the space of networks that integrates both the structure and the order / Disorder and the extra feature such as energy, We can find things that with traditional methods of complexity analysis cannot. 

You may wonder and where complexity connects with the differentiation of climates? Since Merida is a system without bodies of water, its heating dynamics is like a plate under heating, while in coastal cities there are more factors that alter the ambient temperature, in this case the sea and the swamp, which makes it a system more complex. In the middle is Sierra Papacal, which is a city that is between these two extremes, close to the effects of sea breezes but without bodies of water. We think that if Mérida is the least complex and Progreso is the most complex from the point of view of factors that affect the heating-cooling dynamics, Sierra Papacal would be an intermediate. DFA and ApEnt, could not differentiate between Mérida and Sierra Papacal, generating the same magnitude of complexity for these two cities, which cannot be because Sierra Papacal is closer to the effects of sea breezes than Mérida and is also in a intermediate climatic zone between Progreso and Merida according to the Köpen climate classification modified by García. What we find with the networks is that 1) if we can differentiate with NVA and HVA the 3 cities and these differences are statistically significant, so we can assign a measure of complexity across the space of the networks that conventional methods such as DFA or ApEnt do not .

Another great result is that we observe that HVA and NVA extract different information from the dynamics. In the literature, an in-depth analysis of what information is extracted by one algorithm or another has not been reported. What we find is that NVA has an upward or downward trend depending on which topological index is being analyzed when you measure the dynamics from the coast to land inside. On the other hand, HVA always highlights the city that lies between the coast and inland, and that in the allocation of Köpen climate zones Modified by Garcia, is an intermediate climate between wet and dry.

Since the assignment of links for HVA is much more restrictive than NVA because of how the algorithms are constructed, we hypothesize that HVA extracts more subtle information than NVA, so that the first one is much more sensitive and allows us to detect states of transition in complex systems. NVA, on the other hand, is able to extract more general trends from the dynamic. Which will also be addressed using different computational experiments in another article.

In summary, this study is focused on generate the basics for a more comprehensive exploration of the quantification of the complexity of a dynamic. The time series of ambient temperature of different cities was taken as an example since they come from a complex system which has been widely studied and whose results we can rely on to rate the effectiveness of the methods. In particular, the test focused on the statistically significant distinction of the dynamics analyzed by several methods, HVA and NVA being better than DFA, ApEnt.

In addition to the results obtained in this initial experiment, many lines of research were generated that are being worked on or that can be taken up by other researchers in such a way that a different twist is given to the analysis of time series using the mapping to networks, going from being merely descriptive to being an analysis that can measure the complexity of a dynamic in a robust way and can be applied with great certainty in other systems more difficult to understand as physiological signals.

We do not talk about the fact that we are measuring the complexity of a dynamic because it is a definition that must be taken very carefully and the approach of analyzing time series using networks with a complexity approach is just beginning to mature. We have corrected the article (introduction and discussion) in such a way that it can be understood as it is an exploratory analysis of complexity of a dynamics through the “space of networks” applied to the climate, and not in itself a purely climatic analysis.

2. Why are data collected in every two minutes, and then averaged in 10 minutes period, and then analysis done on a monthly basis? It is not clear.

A. This is how the equipment works in such a way that the error between the measurements can be reduced and the magnitude generated as accurately as possible.

3. On page 6, first sentence mentions the bell max at k=40 and 33. How these values can be interpreted? What is the meaning of having k=40 or 33? Please give some feedback.

A. The maximum in the degree distribution represents the number of average neighbors that a node will have. The higher the <k> means there are more connections in the network. A time series without fluctuations (smooth), would be generating a network with a higher connectivity than a series that has variations (rough), as in Figure 6. Mérida has a very stable heating and cooling, while Progreso has one more irregular. This irregularity is reflected in its <k>. The graph shows more links for Merida than for Progreso

4. Why on the first 5 days of My 5, 2015 chosen to use the network?

A. We wanted to show visually how the topology of the time series is reflected in the topology of the network generated for each city, so the reader can appreciate easily how the mapping works. Show each valley and each maximum where it is reflected in the network. For this we look for the month with the least amount of fluctuations caused by rains, excess cloudiness or cold fronts, and that is the month of May, which is the hottest and driest and therefore, generates a very stable dynamic conducive to show the differences

5. On page 7, line 173, it is written that “NVA has more connections then HVA”. What is the advantage/disadvantage of having more connections?

A. The difference in connections is that this is the construction of the algorithm, as shown in Figure 1, NVA allows visibility at any angle, while HVA only visibility horizontally, so the first one will have more links than the second. From the results we have the hypothesis that HVA and NVA work for different applications. HVA can work to detect small variations, which NVA cannot because of the high volume of links. While NVA can work to quantify structures that have a lot of tendency or seasonality and where the structure of the time series is very different. However, these hypotheses should be tested in a computational experimentation which is being worked on for another article.

6. On Figure 5 there is a star shape with 6 points for Merida. How one can interprets these 6 points? Also, there are few connections on the middle. What is the meaning of this?

This can be explained more easily in Figure 6, where it is observed in upper and lower case what each of the points refer to, in lowercase are the maximum temperatures, from maximum to maximum (From lowercase letter to lowercase letter) is the visibility between the decrease and temperature rise, which we will call "U" by the shape. When we have 2 non-consecutive maxima that are linked, for example "b" and "d", we find that it is because the maximum "c" is lower than "b" and "d", allowing these last two "to see each other" Yes and form a link according to the algorithm. That's why you see those lines between the clusters.

7. On page 8 line 203 “my” should be “may”.

Thank you! We already corrected the error in the article

We thank you very much for your time and we look forward to your comments.

Reviewer 3 

Thank you very much for your comments, here are the answers

Reviewer #3: This paper is, I think, most naturally classified within the realm of "topological data analysis" (TDA), an area that I have been aware of for several years, though in my own case, only as an observer not as a participant. I have yet to see a paper in this field that, in my own view, provides a convincing analysis of an applied problem that could not have been achieved by more conventional methods, and this paper does not break that trend. Nevertheless I think the scientific literature should be open to new points of view even if they are not, initially, fully developed, and I feel the current paper should be publishable with some revisions intended mainly to clarify details of the method and to correct some minor errors.

My skepticism about the paper comes down essentially to this: there are established "linear" methods of time series analysis, such as autocorrelation plots, fitting autoregressive and moving-average models, spectral analysis and (if one slightly broadens the scope of the problem) multivariate methods such as principal components analysis and factor analysis, that could have been applied to address the problems in this paper, which essentially comes down to a classification problem distinguishing temperature time series at different locations in Yucatan. So if there is one "big picture" question I would like the authors to address in their revision it is this: why, in the authors' view, are the methods in the present paper superior to these long-established techniques?

Overall judgment: my principal concern about this paper is the one I expressed at the beginning, that the authors don't really explain why they need these particular techniques against far better established methods from the time series literature, but if they can clear up the various ambiguities about what they were actually doing, I would be willing to see this paper published in PLoS ONE.

A. What is pursued in this study is to generate the basics to analyze the dynamics from a more holistic point of view such as its transformation to networks. The objective of this article is not to use NVA and HVA as techniques that replace the ways in which climate is classified, but to analyze the problem from another point of view: the complexity of the dynamics through networks analysis. 

The objective of the article is to show if analyzing the dynamics as networks can generate different insights from those of other techniques such as DFA and ApEnt wich are considered basic techniques to measure complexity. We used climate as a study system, in particular the quantitative differentiation between the dynamics of cities with different geographies but relatively close to each other. We did it in this way since this system is very well known, which allows us to check without a doubt which techniques work best. We found that HVA and NVA can distinguish between geographically close dynamics (Merida vs Sierra Papacal) while DFA and ApEnt do not, which proves the hypothesis that analyzing time series using networks can generate complementary results than traditional techniques.

An approach to characterize dynamic systems is based on their complexity, which can be measured by:

• Structure or Self-Affinity, quantified by fractal dimension analysis, DFA

• Attractor in phase space, quantified by exponents of lyapunov, correlation dimensions

• Order / Disorderer state, studied using entropy and its different definitions (ApEnt, SampEnt)

These 3 ways of analyze complexity and measure it are interdependent and interrelated, however, each one focuses on a main characteristic of dynamics.

We do not contemplate autocorrelation plots, fitting autoregressive and moving-average models, spectral analysis and multivariate methods such as principal components analysis and factor analysis because, until the moment the work was done, these techniques are not used to measure complexity. Certainly the meaning of the article seems to be more a way of classifying and for this you can use these techniques or other machine learning techniques, but as we mentioned in this explanation, the meaning of the article is to establish the basis for measuring the complexity of the signals from of the analysis of the generated networks.

Lacasa in his article showed that the mapping with NVA and HVA can extract the structural and order / disorder characteristics of a dynamic. When the network is obtained, its connectivity can be characterized to analyze its structure (similar to self-affinity and fractal techniques) and we can analyze its order / disorder through the entropy of its degree distribution (similar to SampEnt and ApEnt). For the phase state attractor a simile has not yet been found, which would be a future job.

When we test the methods, we find that if we analyze the dynamics from the point of view of their structure / self-affinity (DFA) or by order / disorder (ApEnt), these methods could not differentiate between climates (Merida and Sierra Papacal), while that HVA and NVA can, which is a great INITIAL result, which proves to us the hypothesis that when analyzing the dynamics through the space of networks that integrates both the structure and the order / Disorder and the extra feature such as energy, We can find things that with traditional methods of complexity analysis cannot. 

You may wonder and where complexity connects with the differentiation of climates? Since Mrida is a system without bodies of water, its heating dynamics is like a plate under heating, while in coastal cities there are more factors that alter the ambient temperature, in this case the sea and the swamp, which makes it a system more complex. In the middle is Sierra Papacal, which is a city that is between these two extremes, close to the effects of sea breezes but without bodies of water. We think that if Mérida is the least complex and Progreso is the most complex from the point of view of factors that affect the heating-cooling dynamics, Sierra Papacal would be an intermediate. DFA and ApEnt, could not differentiate between Mérida and Sierra Papacal, generating the same magnitude of complexity for these two cities, which cannot be because Sierra Papacal is closer to the effects of sea breezes than Mérida and is also in a intermediate climatic zone between Progreso and Merida according to the Köpen climate classification modified by García. What we find with the networks is that 1) if we can differentiate with NVA and HVA the 3 cities and these differences are statistically significant, so we can assign a measure of complexity across the space of the networks that conventional methods such as DFA or ApEnt do not .

Another great result is that we observe that HVA and NVA extract different information from the dynamics. In the literature, an in-depth analysis of what information is extracted by one algorithm or another has not been reported. What we find is that NVA has an upward or downward trend depending on which topological index is being analyzed when you measure the dynamics from the coast to land inside. On the other hand, HVA always highlights the city that lies between the coast and inland, and that in the allocation of Köpen climate zones Modified by Garcia, is an intermediate climate between wet and dry.

Since the assignment of links for HVA is much more restrictive than NVA because of how the algorithms are constructed, we hypothesize that HVA extracts more subtle information than NVA, so that the first one is much more sensitive and allows us to detect states of transition in complex systems. NVA, on the other hand, is able to extract more general trends from the dynamic. Which will also be addressed using different computational experiments in another article.

In summary, this study is focused on generate the basics for a more comprehensive exploration of the quantification of the complexity of a dynamic. The time series of ambient temperature of different cities was taken as an example since they come from a complex system which has been widely studied and whose results we can rely on to rate the effectiveness of the methods. In particular, the test focused on the statistically significant distinction of the dynamics analyzed by several methods, HVA and NVA being better than DFA, ApEnt.

In addition to the results obtained in this initial experiment, many lines of research were generated that are being worked on or that can be taken up by other researchers in such a way that a different twist is given to the analysis of time series using the mapping to networks, going from being merely descriptive to being an analysis that can measure the complexity of a dynamic in a robust way and can be applied with great certainty in other systems more difficult to understand as physiological signals.

We do not talk about the fact that we are measuring the complexity of a dynamic because it is a definition that must be taken very carefully and the approach of analyzing time series using networks with a complexity approach is just beginning to mature. We have corrected the article (introduction and discussion) in such a way that it can be understood as it is an exploratory analysis of complexity of a dynamics through the “space of networks” applied to the climate, and not in itself a purely climatic analysis.

Besides, in the research group we are not familiar with ADD techniques. At this time of review we review some articles and believe that it is a great tool and is definitely in the same direction, analyze the "form" of the data from a topological point of view, only that we do it through networks. We believe that this approach together with ADD can make a great synergy to be able to explore more particularities of the dynamics. Without a doubt there is much to explore, we thank you very much for sharing the TDA

Page 1, bottom: climate or weather? It seems to me this paper is primarily addressing daily weather patterns in different regions of Yucatan, and I don't see any implication for long-term trends (e.g. whether trends are greater on the coast than inland) following directly from this analysis. Terminology is important, and so is thinking about the broader implications of your work, so if you do see such implications, I would encourage you to develop them.

A. Thank you very much for the comment, we have corrected the vocabulary in the article

Page 2, line 20, DFA is introduced here but not explained, whereas later you write Detrended Fluctuation Analysis. My advice would be to spell out an acronym the first time you use it, but thereafter, once the meaning is established, writing DFA (and other acronyms used in this paper) would be fine

A. Corrected in the article.

Page 2, line 37: the inserted word "it" is redundant. (But my broader concern is not with minor linguistic detail but the broader implication of this sentence: it seems to me you have not considered well-established time series techniques among the "other techniques" that you discuss.)

A. Corrected in the article.

Page 2, line 46: Natural not NAtural

A. Corrected in the article.

Page 2, lines 47-49: I would encourage the authors to be careful about consistency of notation. Why introduce the time series as y_t and then immediately switch to y_i and y_j?

A. t is used at the beginning because a set of numbers with N elements "t" is being introduced. I and j are used to say that they are any t elements. However, if you suggest that we change the notation since the sense of the algorithm can be misunderstood, we will gladly change it

Page 2, equation (1.1): there is something mysterious about how this formula appears and the way it is depicted in figure 1. On the face of it, temperature time series have both positive and negative values and there is nothing special about "zero" whether they are being measure in degrees Fahrenheit or Celsius (of course zero Kelvin does have a special physical interpretation, but weather time series never get anywhere close to that boundary). Formula (1) seems a little odd because if you reversed the signs the condition would change, and correspondingly, the picture in figure 1 would look different if you chose a different base value for zero. Possibly this has been previously discussed in the previous literature on these techniques, but it seems to me the classification would change if you reversed all the signs, and that with temperatures, somehow this should not happen. Any comment on these issues?

A. What the algorithm does (equation 1.1) is to be comparing slopes between the points, so if one of them is negative it does not affect. However, since the algorithm is invariant before all the transformations shown in the figure below, if there were negative values, what could be done is to “upload” the entire time series in such a way that they were not negative and the time series can be transformed to a network

https://www.pnas.org/content/105/13/4972

Page 3, lines 72 and 76: should be no indentation (if the manuscript was prepared using Latex then this is a common Latex error)

A. Corrected in the article.

Page 4, top: please include references to K\"oppen and Garc\\'ia

A. Corrected in the article.

http://www.igeograf.unam.mx/sigg/utilidades/docs/pdfs/publicaciones/geo_siglo21/serie_lib/modific_al_sis.pdf

Page 4, around line 90: this is another general point of presentation that other reviewers might express differently, but my advice is that for methods that you actually use in your analysis, that here include DFA and ApEnt, you should provide enough information in the form of formulas or explicit references (one or the other) so that the reader who wants to repeat your analysis can reproduce it if so desired, whereas for other methods that you only mention in passing, such as Shannon Entropy, it's not necessary to be so explicit.

A. Both techniques are referenced so that people can go see how these parameters are calculated. The article would become very long if it was included how it was calculated.

Page 4, around line 95: please clarify exactly how you are computing the t-test. My first reaction was that since you are looking at correlated time series, any use of t-tests or similar methods should include a correction for autocorrelation, but then I realized that if you are using a t-test to compare measures computed from widely separated blocks of the time series, maybe this kind of correction is not necessary. In nay case, I feel this point deserves some explicit discussion, i.e. either include a correction for autocorrelation or explain why it is not necessary.

A. Corrected in the article. A t-test is done to know if the results obtained are statistically different or not for each measure of the network or for the DFA, ApEnt, the mean or the standard deviation. We want to do this to know if the parameter we are using to measure can distinguish between the two cities. Population A will be composed of the calculation of each month, that is 24 elements, also population B. There are no autocorrelation problems because we are not comparing the time series itself, but the measurements of the networks or DFA or ApEn per month of each population, which can be considered as independent events from each other.

Page 4, line 105: when you say that Merida shows a "significant" difference (but Sierra Papacal, for example, does not) you should state exactly what numerical measure you are using to judge what is "significant".

A. Corrected in the article. (p-valule <0.05, 95 \\% of confidence)

Page 4, lines 107-109: I am still looking for a meteorological interpretation of these results. It seems to be that, very broadly, what you are measuring is smoothness of the diurnal variation, and it is a general phenomenon that temperatures on the coast show less diurnal variation that those inland. Is this what is going on, or should we be thinking about some more complex interpretation?

A. It was corrected in the article. Combining the results of Table 1 and 2 we can say that there are only statistically significant differences between Merida and Progreso using the mean and standard temperature deviation, as well as ApEnt. This means that Mérida has the most dispersed data of the average than Progreso, this is due to the buffering by the sea. Regarding the interpretation of the ApEnt, it can be said that the dynamics are more random in Progreso than in Mérida and this is due to the waves of fresh air that come from the ocean caused by the difference in temperatures between the sea and the land.

Page 4, line 110: "very similar values". Same comment as the one about significance in Merida: I would like to know exactly what the numerical values were and how you judged that they either were or were not significantly different for the different cities.

A. It was corrected in the article

Page 4, line 117: same comment as above about the "paired t-test": Please explain in a little more detail about what this was and whether autocorrelation is an issue in the way the t-test is calculated

A. Corrected in the article. A t-test is done to know if the results obtained are statistically different or not for each measure of the network or for the DFA, ApEnt, the mean or the standard deviation. We want to do this to know if the parameter we are using to measure can distinguish between the two cities. Population A will be composed of the calculation of each month, that is 24 elements, also population B. There are no autocorrelation problems because we are not comparing the time series itself, but the measurements of the networks or DFA or ApEn per month of each population, which generate independent events from each other.

Page 4, line 119, "all measures except the mean and DFA". Slightly strange wording here: it would be more straightforward to list the measures that were different than those that were not, in other words, say that SD and ApEnt showed a significant difference between these two cities

A. Corrected in the article 

Page 5, line 130: magnitude

A. Corrected in the article

Page 6, lines 152-154: as with earlier comments about how you assess differences among cities, I would like you to be a little more explicit here, how exactly are you making these judgments.

A. Corrected in the article

Page 7, caption to Figure 5: I have to say that the interpretation of the plots is becoming a little harder for me as the paper goes on. Figure 5 has a pleasing geometric appearance but I am not at all sure how to interpret it. I think the relevant scientific question is this: are the results dependent on the specific choice of a 5-day window or are the authors in a position to state that there are some general patterns emerging in these plots that are invariant to irrelevant details such a which specific 5 days we chose for the analysis? I didn't see that question discussed.

A. Corrected in the article. We wanted to show visually how the topology of the time series is reflected in the topology of the network generated for each city, so the reader can appreciate easily how the mapping works. Show each valley and each maximum where it is reflected in the network. For this we look for the month with the least amount of fluctuations caused by rains, excess cloudiness or cold fronts, and that is the month of May, which is the hottest and driest and therefore, generates a very stable dynamic conducive to show the differences

Page 7, caption ot Figure 6: please check the wording of this caption for typos (repeated "it is") and Progreso is misspelled at one point.

A. Corrected in the article

Page 9, lines 255-257, English please!

A. Corrected in the article

Figure 4: could you please explain why the density plot appears to go below zero at around k=2?

A. Fixed in article. It was a mistake in the edition of the figure

On data availability: Note that the time series are in ten-minute intervals and the standard sources for meteorological data (e.g. NOAA or NASA in the USA) only list daily or monthly time series. I think the authors could give more explicit detail about the original source for the data (e.g. did they come from the Mexican meteorological service?) and how the data were prepared for the present application - if the original source of the data was public then there is no issue about data confidentiality.

A. The measuring equipment is owned by the CINVESTAV Department of Marine Resources, which is responsible for collecting the data and maintaining the equipment. The data acquisition process described in the article is the way in which the equipment operates by default. 

The data will be public

Thank you very much for your time and we look forward to your comments. It was very helpful to clarify the real aim of the paper.

---

## [Decision Letter · Decision Letter 1]

9 Oct 2019

PONE-D-19-14940R1

Temperature Time series analysis at Yucatan using Natural and Horizontal Visibility Algorithms

PLOS ONE

Dear Mr Rosales Perez,

Thank you for submitting your manuscript to PLOS ONE. After careful consideration, we feel that it has merit but does not fully meet PLOS ONE’s publication criteria as it currently stands. Therefore, we invite you to submit a revised version of the manuscript that addresses the points raised during the review process.

The reviewers agree that the paper has much improved from the initial version, and a revised version could be published in PLOS ONE. In particular, I encourage the authors to properly address all the comments by the reviewers, for what concerns the presentation of the paper. A thorough proofreading is necessary to fix all the typos and grammatical issues.

We would appreciate receiving your revised manuscript by Nov 23 2019 11:59PM. To enhance the reproducibility of your results, we recommend that if applicable you deposit your laboratory protocols in protocols.io, where a protocol can be assigned its own identifier (DOI) such that it can be cited independently in the future. For instructions see: http://journals.plos.org/plosone/s/submission-guidelines#loc-laboratory-protocols

We look forward to receiving your revised manuscript.

Kind regards,

Marco Lippi

Academic Editor

PLOS ONE

Additional Editor Comments (if provided):

The authors should revise the paper according to the comments by the reviewers. In particular, I suggest the authors to provide a very careful proofreading of the paper to fix all the typos and grammatical issues raised by the reviewers.

Reviewers' comments:

Reviewer's Responses to Questions

**Comments to the Author**

1. If the authors have adequately addressed your comments raised in a previous round of review and you feel that this manuscript is now acceptable for publication, you may indicate that here to bypass the “Comments to the Author” section, enter your conflict of interest statement in the “Confidential to Editor” section, and submit your "Accept" recommendation.

Reviewer #2: All comments have been addressed

Reviewer #3: (No Response)

2. Is the manuscript technically sound, and do the data support the conclusions?

Reviewer #2: Yes

Reviewer #3: (No Response)

3. Has the statistical analysis been performed appropriately and rigorously? 

Reviewer #2: Yes

Reviewer #3: Yes

4. Have the authors made all data underlying the findings in their manuscript fully available?

Reviewer #2: Yes

Reviewer #3: Yes

5. Is the manuscript presented in an intelligible fashion and written in standard English?

Reviewer #2: No

Reviewer #3: Yes

6. Review Comments to the Author

Reviewer #2: Thank you very much for the detailed answers to all issues that is raised by the reviewers. I have a small question that bothers me. Since you use statistical testing procedures, are there any assumptions like normally distributed variables? Please correct some typos that you add in this revision. Especially p-value<0.05 parts.

Reviewer #3: The authors have responded in detail to all the reviews and have improved the paper. My doubts about the ultimate merits of the method remain, but the authors have done a decent job of explaining what they are doing and why. I think the paper could go forwards at this point, but it does need at least one more revision.

I went through the paper again, highlighting all the places where there were typos or minor grammatical errors (not fatal to the paper, but irritating nonetheless) and also a few where I thought the interpretation was wrong or questionable. In particular, I noted a number of places (especially concerning the various pairwise tests presented) where statements in the text did not exactly correspond to the evidence in the tables. Some latitude of interpretation is allowable in a paper of this nature, but the authors should still take care to avoid a direct contradiction between the text and the tables or figures. And I still think Figures 5 and 6 are very hard to interpret and would recommend these be omitted (except perhaps for the time series plots in Figure 6 which do help to explain why the time series plot for Merida appears smoother than that for Progreso).

Specific comments:

Abstract: insert "the" before "Natural Visibility Algorithm"

Introduction line 5: "what" should be "which"?

End of same line: should "dynamic" be "dynamical system"?

Page 2 line 21: should it be "Approximate Entropy or Sample Entropy"? (assuming that these are two different things and not two different names for the same thing)

Line 34: I assume you meant "associated"

Line 42: "approach"

page 3, line 67: this seems awkwardly worded. Should

"average of neighbors that have the nodes in the network"

be

"average number of neighbors over all nodes in the network"?

line 80: suggest "to characterize the network, other measures have been suggested such as Shannon's Entropy ... "

page 4, line 118 and a number of subsequent places: here I think you meant to say p-value < 0.05 but the < appears in my pdf as an upside down "!"; this could be a glitch in the pdf conversion process. Please do a global search for errors of this nature. Also, here and a number of subsequent places you wrote "p-valule" where obviously it should be p-value (the next instance is line 125)

line 115: independent

line 124: Here I have a slightly more substantial point. You are saying that Merida shows a significant difference (in mean temperature) from the coastal cities whereas SierraP does not. But in fact, the gap from Merida to Sisal is smaller than the gap from Sisal to Progreso (the third largest among these five). So I think some further clarification is needed exactly what you are comparing with what here.

line 139: you didn't respond to my previous comment here. It still seems odd to me to say "all measures except..." rather than simply list the measures that do show a significant difference (between SierraP and Progreso)

lines 140-141: here is one place where what you say does not correspond to what is in the table. For example, the SD is significantly different between Sisal and Progreso, and the mean is significantly different between Merida and SierraP, in both cases contradicting what is in the text. (However, in each case the p-value is of the order 0.02, which may not be significant when multiple testing considerations are taken into account. Obviously, in this sort of context a p-value of 0.02 is nothing like one of 0.00001 say, as in some of the SD differences. If the authors made it explicit that this is the kind of comparison they are drawing, then I would say, fair enough.)

line 163: this is another place where the text does not exactly correspond to the table. In particular, geodesic does distinguish between Merida-SierraP and assortativity distinguishes between SierraP-Progreso (though, again, with p-values >0.01 which could be interpreted as "not too significant" in the context of many simultaneous tests being performed pairwise among these cities)

Table 4, line 2: "SierraP"

lies 174 and 178: again, I am having trouble with "higher the ... means" and "reflected in its ..." where in each case the "..." has been rendered as something unintelligible, though this is probably again an artifact of the pdf conversion

line 187: p-value < 0.05 (again)

lines 191-192: another sentence where the assertion does not appear to be true (note also "SierraP" in line 3 of Table 6)

line 203: plotting

Figures 5 and 6: I still have a hard time understanding the circular plots. Apart from the visual interpretation (or lack on it), I don't understand why the authors are trying to infer general characteristics of the time series from a sample of only five days (where the authors even admit that the five days were chosen to illustrate a specific point). I would just omit these two plots to avoid confusing the reader.

line 288: can be (two separate words)

line 289: should "bases" have been "basis"?

Data availability: the authors have provided a zip directory which appears to resolve this issue.

Overall verdict: the authors should go through all the above points and make specific corrections where needed, and I would also recommend they give the whole manuscript a very careful read-through to look for other similar points that I may not have spotted on my admittedly rather quick reading of the paper. If they can make such corrections, however, I feel the next version of the paper ought to be acceptable.

7. PLOS authors have the option to publish the peer review history of their article (what does this mean?). If published, this will include your full peer review and any attached files.

Reviewer #2: No

Reviewer #3: No

---

## [Author Response · Author response to Decision Letter 1]

16 Nov 2019

Dear Reviewers and Editor

We appreciate your comments and suggestions. They helped us a lot to improve the article and also to be able to give a clearer understanding of the idea that we want to communicate to the readers. We hope that with these corrections the points we want to release will become clearer. We thank you for your time and patience.

About the main observation that you mentioned about the writing, we decided to send the document to a writing style correction agency to ensure the quality of the manuscript. We attach the certificate that was given to us along with the corrections.

The letter is organized as follows: First the response to the editor and then the response to the reviewers. Each answer is a letter for each one. We include the editor's comment (in bold) and the answer begins with an A.

Once again we thank you for your support. We look forward to your comments

Reviewer 2

Thank you very much for the detailed answers to all issues that is raised by the reviewers. I have a small question that bothers me. Since you use statistical testing procedures, are there any assumptions like normally distributed variables? Please correct some typos that you add in this revision. Especially p-value<0.05 parts.

A. We thank you very much for your comments, in fact to apply this test the data has to be independent and normal. The independence of the data is obtained since the measurements were made in each city with its own measuring instrument and there is no relationship from one city to another in the measurement process.

For the normality of the distribution of the data a statistical test was made. Usually, Kolmogorov-Smirnof test is used, but since there are only 21 data, Shaphiro-Wilk is used, which applies to samples of 50 records or less. We obtain the following values of p. The green color indicates that p> 0.05 wich indicates normally distributed data. If p <0.05 we have non-normal data (Yellow color).

 Measurement Merida Progreso Sisal Telchac SierraP

General Media 0.00330591 0.00054404 0.0044705 0.00869545 0.00244482

 DevStd 0.12769559 0.03318864 0.22441483 0.02037118 0.98413329

 ApEnt 0.73681198 0.57449912 0.64037782 0.37098163 0.22432368

 DFA 0.25759864 0.77971655 0.66232862 0.62602547 0.48277831

NVA mean deg 0.31455408 0.34683294 0.15390243 0.3379569 0.54944949

 geodescics 0.12160723 0.16405199 0.15892265 0.15281428 6.50E-05

 clustering 0.58601159 0.09595623 0.22182491 0.00849342 0.85359038

 assortativity 0.08866683 0.69267346 0.23608482 0.7582951 0.74569767

 Laplacian Ener 0.08572239 0.79764814 0.39048238 0.00037081 0.05938504

 Shannon Ent 0.24241821 0.07347954 0.06860002 0.22738198 0.55462299

HVA mean deg 0.53432317 0.25306173 0.20985021 1.46E-05 0.37940119

 geodescics 0.07481793 0.02790849 0.24040644 0.19073564 0.6971402

 clustering 0.40871383 0.43748458 0.03630906 1.18E-07 0.66769125

 assortativity 0.99948168 0.33598857 0.53141366 0.00404848 0.11568569

 Laplacian Ener 0.87650306 0.2982707 0.30343033 1.80E-09 0.18310141

 Shannon Ent 0.65826936 0.13908803 0.39783225 2.38E-06 0.05017746

Given this analysis, we have decided to include only cities and measurements that do meet normal data (green in rows and columns). So we will only stay with:

Cities: Merida, Progreso, Sierra Papacal and Sisal

Measurements: ApEnt and DFA, Mean Degree, Assortativity, Laplacian Energy, Shannon Entropy. 

This ensures us to continue comparing all measurements with all forms of transformation algorithm and among all cities since they all belong to normal distributions.

We included mean and standard deviation of the temperature in Table 1 but only as a reference to the readers, but these were not included in the t-test analysis.

Also, without the variables and measurements we took away, the paper is much more clear. Thank you for this important observation.

Reviewer 3

Thanks for the feedback. As we mentioned at the beginning, all the errors and misunderstandings in writing were corrected, as well as the typos and symbols that were not correctly reflected in the PDF. Next we add the concrete answers to the corrections that are not related to typos, writing or contradictions between what is shown in the tables and what is commented in the manuscript:

Figures 5 and 6: I still have a hard time understanding the circular plots. Apart from the visual interpretation (or lack on it), I don't understand why the authors are trying to infer general characteristics of the time series from a sample of only five days (where the authors even admit that the five days were chosen to illustrate a specific point). I would just omit these two plots to avoid confusing the reader.

A. The entire “Comparing Extreme cities” subsection was omitted because we realized that it only confuses the reader instead of generating more clarity or information that supports the main conclusions. The images were also removed leaving only the time series to show the part we need to be clear in the readers.

---

## [Editor Report · Decision Letter 2]

4 Dec 2019

Temperature Time series analysis at Yucatan using Natural and Horizontal Visibility Algorithms

PONE-D-19-14940R2

Dear Dr. Rosales Perez,

We are pleased to inform you that your manuscript has been judged scientifically suitable for publication and will be formally accepted for publication once it complies with all outstanding technical requirements.

With kind regards,

Marco Lippi

Academic Editor

PLOS ONE
---

## [Editor Report · Acceptance letter]

11 Dec 2019

PONE-D-19-14940R2 

Temperature Time series analysis at Yucatan using Natural and Horizontal Visibility Algorithms 

Dear Dr. Rosales Perez:

I am pleased to inform you that your manuscript has been deemed suitable for publication in PLOS ONE. Congratulations! Your manuscript is now with our production department. 

With kind regards,

on behalf of

Dr. Marco Lippi 

Academic Editor

PLOS ONE